# Low-force pulse switching of ferroelectric polarization enabled by imprint field

Yuchao Zhang [1], Shanzheng Du[1], Xiaochi Liu[1], Yahua Yuan[1], Yumei Jing[1], Tian Tian [2] ✉, Junhao Chu[2], Fei Xue [3,4] ✉, Kai Chang[3] & Jian Sun [1,5] ✉

Beyond conventional electrical modulation, flexoelectricity enables mechanical control of ferroelectric polarizations, offering a pathway for tactile-responsive ferroelectric systems. However, mechanical polarization switching typically requires substantial static threshold forces to overcome the significant energy barrier, resulting in material fatigue and slow response that compromises reliability and hinders practical applications. In this work, we address these challenges by introducing an imprint field through asymmetric electrostatic boundary design with distinct work functions. This built-in electric field stabilizes the energy landscape, effectively lowering the polarization switching barrier. Subsequently, nonvolatile polarization switching with a low threshold force of 12 nN·nm⁻¹ is achieved in $CuInP_2S_6$ without material damage. Surpassing the limitations of slow static force controls, our work marks the first experimental demonstration of fast mechanical control of polarization switching with 4 millisecond-long low force pulses. To further highlight the potential of this rapid, low-force mechanical control, we propose a van der Waals heterostructured mechanically gated transistor with asymmetric electrostatic boundary, which exhibits gate force pulses-controlled multi-level, nonvolatile conductance states. Our findings establish a paradigm for next-generation ferroelectric electronics that integrate responsiveness to mechanical stimuli.

Ferroelectric materials have long been a cornerstone of modern electronics due to their unique ability to exhibit spontaneous electric polarization that can be reversed by an external electric field[1,2]. Beyond electrical control, ferroelectric polarization can be manipulated using mechanical force through the flexoelectric effect[3–7]. This effect arises from a non-uniform mechanical deformation, or strain gradient, which generates an internal flexoelectric field capable of realigning ferroelectric domains[8–10]. This approach offers an additional way to control ferroelectric properties without relying solely on electric fields[11–13], thereby enabling the possibility of emerging ferroelectric devices with tactile sensitivity[14,15]. Such a mechanism not only broadens the scope of

ferroelectric applications but also paves the way for emerging devices with tactile sensitivity, where mechanical stimuli can directly modulate electronic responses[16–19].

Despite its potential, the practical implementation of flexoelectricity in functional devices remains challenging, particularly in achieving efficient and low-threshold polarization switching. One of the primary obstacles is the high threshold force required to induce polarization switching. As the flexoelectric effect is directly proportional to the strain gradient[20,21], a sufficiently high force is required to generate a large enough strain gradient to modulate the ferroelectric domains[22]. However, applying such high forces locally can lead to

[1]School of Physics, Central South University, Changsha 410083, China. [2]Department of Materials Science, Fudan University, Shanghai 200433, China. [3]Center for Quantum Matter, School of Physics, Zhejiang University, Hangzhou 310058, China. [4]ZJU-Hangzhou Global Scientific and Technological Innovation Center, Zhejiang University, Hangzhou 311215, China. [5]State Key Laboratory of Powder Metallurgy, Central South University, Changsha 410083, China. ✉e-mail: 2111030057@m.fudan.edu.cn; xuef@zju.edu.cn; jian.sun@csu.edu.cn

irreversible material degradation, including plastic deformation[23] or ion migration[24–27], which degrades the structural integrity and functionality of the ferroelectric materials.

Recent advancements have demonstrated that nanoscale systems offer a promising avenue for addressing these challenges. Due to the pronounced strain gradients inherent in nanoscale geometries[28,29], flexoelectricity-induced polarization switching has been successfully observed in ultrathin ferroelectric films[30–32]. Moreover, strategies such as strain engineering and domain engineering have been developed to reduce the threshold force required for mechanical polarization switching[33,34]. These approaches utilize controlled strain distributions and optimized domain configurations to enhance the efficiency of flexoelectric effects while minimizing mechanical stress. However, despite these advancements, integrating such methods into practical device architectures, particularly the heterostructured transistor configuration[35], remains technically challenging. The complexity of fabricating and controlling nanoscale strain gradients, coupled with the need for precise domain engineering, poses significant hurdles for large-scale implementation.

More recently, the discovery of two-dimensional layered ferroelectrics, i.e., $CuInP_2S_6$ (CIPS)[36,37], has provided an opportunity to explore the flexoelectric effect and its applications[38–40]. The layered structure and ultra-low thickness, combined with its inherent flexibility and strong electromechanical coupling, position CIPS as a promising candidate for developing mechanically tunable ferroelectric devices. For instance, CIPS can be easily suspended or placed on an elastic polymer to significantly enhance the strain gradient under applied mechanical force, therefore enabling polarization switching at lower force[41,42]. However, such structures pose challenges for designing functional ferroelectric devices on the rigid substrates and the integration with other functional devices.

In this work, we propose a strategy to lower the threshold force for flexoelectricity-mediated polarization switching by implementing a controlled imprint field via asymmetric boundary design. This configuration stabilizes an imbalanced energy landscape, therefore effectively reducing the switching barrier. Using this approach, we demonstrate effective polarization switching in CIPS with 4 ms-long low-force pulses of 600 nN, achieving a record low threshold force per thickness of 12 nN·nm⁻¹. In contrast, without the imprint field, a significantly higher force of >20 nN·nm⁻¹ not only fails to induce polarization switching but also causes noticeable damage to CIPS. The asymmetric contact configuration is fully compatible with electronic device architectures. As an application demonstration, we developed a mechanically-gated ferroelectric transistor exhibiting nonvolatile, multi-level conductance controlled by rapid low-force mechanical stimuli. This work offers a pathway for effectively controlling polarization in low-dimensional ferroelectrics via the flexoelectric effect and provides a foundation for developing mechanically controlled ferroelectric devices.

## Results

### Switching barrier lowering by imprint field

We first explain the strategy for enabling low-force polarization switching by introducing an imprint field via asymmetric electrostatic boundary conditions in ferroelectric materials. As known, the energy landscape of ferroelectric materials, characterized by two degenerate polarization states, can be described using the Landau double-well model as plotted in Fig. 1a. In the presence of a strain gradient, flexoelectricity introduces a polar bias, which tilts the energy landscape and creates an imbalance between the two potential wells. To achieve polarization switching, the mechanical stimulus must tilt the potential sufficiently to destabilize the current polarization state, overcoming the energy barrier $\Delta E_1$ and driving the system into the opposite polarization state with lower energy. Therefore, to achieve polarization switching with lower mechanical force, it is necessary to reduce

the height of this energy barrier. It is important to note that the strain gradient does not symmetrically tilt the energy landscape in both directions. Instead, it always drives the polarization toward a specific state, resulting in unidirectional switching. An imprint field in a ferroelectric material is an internal electric field that biases the polarization state, favoring one direction over the other. In ferroelectric devices, imprint fields can have both detrimental and beneficial effects. On one hand, they can lead to asymmetric switching behavior and retention loss in the energetically disfavored polarization state. On the other hand, they can be engineered to guide a desired polarization direction by reducing the energy barrier. In terms of flexoelectric-induced polarization switching, an imprint field aligned with the flexoelectric field can effectively lower the energy barrier for one polarization direction ($\Delta E_2 < \Delta E_1$), enabling switching at reduced mechanical force. Such imprint field can arise from structural asymmetries, e.g., differences in work function between the top and bottom electrodes. Achieving precise control of the imprint field is complex, as it requires comprehensive optimization of both the electrode material properties and the ferroelectric thickness, considering the interplay of work function differences, charge screening, interfacial chemistry, and size-dependent electrostatic effects. Nevertheless, by contacting the ferroelectric surfaces with selected materials, an imprint field can be introduced to facilitate flexoelectric polarization switching.

This scenario is validated using 50 nm-thick CIPS samples placed on titanium (Ti) with a low work function of 4.33 eV, as illustrated in Fig. 1b. Mechanical force is applied to the surface of the CIPS using a platinum (Pt)-coated scanning probe, where the Pt has a high work function of 5.56 eV. In this configuration, when the probe makes hard contact with the CIPS, it can induce flexoelectricity due to the uneven distribution of deformation caused by the tip[43]. An additional CIPS sample is prepared on gold (Au) with a work function of ~5.20 eV for comparison. CIPS possesses a band gap of 2.70 eV and an electron affinity of 3.70 eV[44]. The Fermi energy of CIPS is experimentally determined by measuring the surface potential difference between CIPS and Ti (Au) using Kelvin probe force microscopy (KPFM)[45,46]. As shown in Fig. 1c, the surface potential of CIPS is ~360 meV below that of Au and ~460 meV above that of Ti, indicating a Fermi energy of ~4.80 eV. Subsequently, the energy-level alignments before and after the contact of the Pt tip on Au and Ti can be illustrated as Fig. 1d. Due to the higher work function of Pt compared to CIPS, electron transfer occurs from the CIPS interface to Pt upon contact. A similar process occurs at the CIPS/Au interface, though the work function difference is smaller, leading to fewer accumulated holes compared to the Pt/CIPS interface. As a result, a weaker imprint field, $E_{imp}$, is introduced in CIPS pointing from Pt to Au. In contrast, when Ti is in contact, its lower work function causes electron transfer from Ti to CIPS, resulting in significant electron accumulation and therefore a stronger $E_{imp}$ pointing from Pt tip to Ti. Consequently, this leads to a significantly tilted energy landscape with a low energy barrier for downward switching of polarization.

### Imprint field modulated switching of polarization

We confirm the out-of-plane ferroelectricity of CIPS using piezoresponse force microscopy (PFM) measurements[47]. Supplementary Fig. 1 shows out-of-plane PFM phase and amplitude images of CIPS, revealing distinct ferroelectric domains with spontaneous polarization. Polarization switching is further verified by applying a bias voltage to the CIPS surface using a Pt probe. As shown in Fig. 2a, a box-in-box pattern with reversed polarizations exhibits a 180° phase difference, written using ±4 V DC voltage. This confirms the reversible nature of polarization in CIPS under an electric field.

Owing to the titled energy landscape, the asymmetric contact configuration of Pt/CIPS/Ti also plays a role in facilitating downward polarization switching at lower voltages. To verify this, we measured local phase hysteresis loops on both Ti and Au substrates. As illustrated

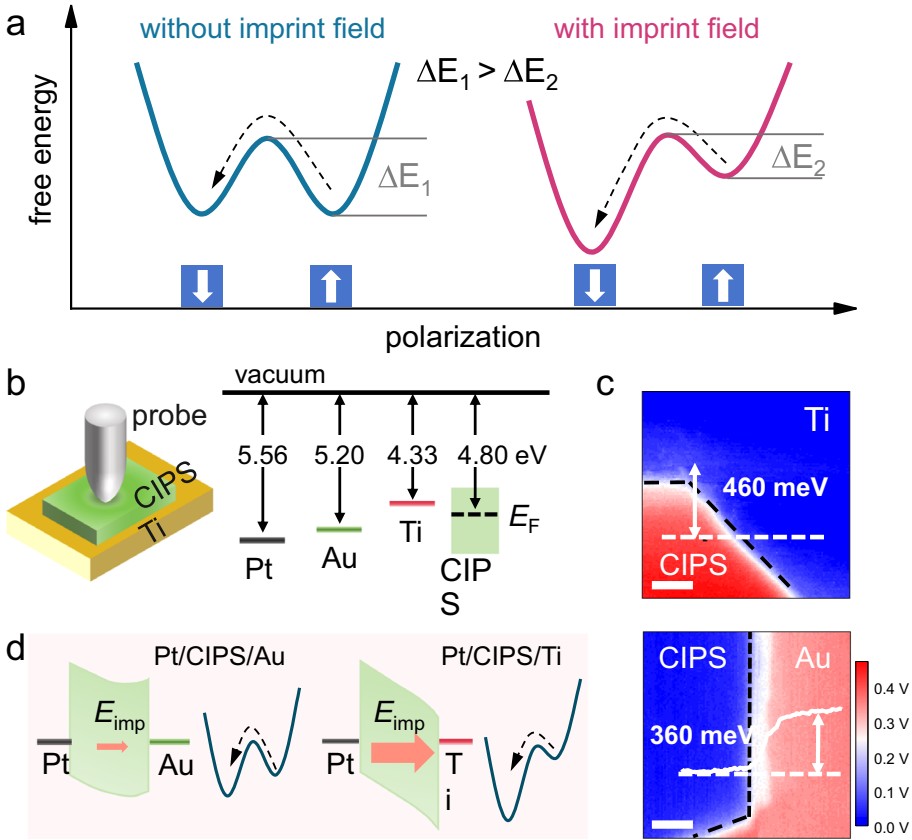

**Fig. 1 | Lowering switching barrier for CIPS by imprint field. a** Schematic illustrating the strategy to reduce the flexoelectric switching force by inducing imprint field. The double-well potential of the Landau free energy is shown for a ferroelectric under without and with internal imprint field, with the energy barrier for polarization switching reduced with imprint field. **b** Schematic of a CIPS placed on Ti, with a top scanning platinum probe applying mechanical force. The energy diagram depicts the alignment of energy levels between the Pt, CIPS, and Ti (Au). **c** Kelvin probe force microscopy surface potential images of CIPS on Ti and Au, respectively. Scale bar: 200 nm. Insets show surface potential profiles along the cutting lines. **d** Energy band diagrams demonstrating the significant imprint field in the Pt/CIPS/Ti structure, and the lower energy barrier for polarization switching in CIPS compared to the Pt/CIPS/Au structure. Scale bar: 200 nm.

in Fig. 2b, the hysteresis loop on Ti exhibits a pronounced negative shift along the voltage axis, indicating the presence of ferroelectric imprint effect[48,49]. This shift reflects a tilted energy landscape, which lowers the energy barrier for polarization switching. Specifically, the threshold voltage $V_{th}$ for downward switching on Ti is reduced to 2.33 V, compared to 2.70 V for CIPS/Au. This reduction highlights the effectiveness of the asymmetric contact design in enhancing switching efficiency, aligning with the theoretical framework of the imprint field modulated energy landscape discussed earlier.

To quantitatively evaluate the flexoelectric effect, we conducted finite element analysis to calculate the flexoelectric potentials generated on a 50 nm-thick CIPS layer under applied mechanical force. As shown in Fig. 2c, a tip force of 600 nN generates a surface flexoelectric potential of 2.39 V, which exceeds the 2.33 V threshold voltage observed for CIPS/Ti in the hysteresis loop measurements. Additionally, we examined the flexoelectric potential generated by the tip forces of 800 nN and 1000 nN, as plotted in Fig. 2d. Although the flexoelectric surface potential increases to 2.65 V under 1000 nN, it remains below the 2.70 V threshold for CIPS/Au. These results confirm that the flexoelectric effect, with assistance from asymmetric boundary design, is sufficient to induce polarization switching at significantly lowered forces.

### Fast force pulse-controlled polarization switching
The low-force-enabled polarization switching in CIPS/Ti was experimentally verified by applying fast force pulses of 4 ms using a Pt probe. Prior to mechanical stimulation, a fast pulse of negative voltage of −4 V was applied to initialize the measured area to an upward polarization state while preventing material damage from ferroionic effect. PFM phase images were acquired before and after the application of mechanical forces to directly visualize the polarization changes as shown in Fig. 2e. A single 4 ms-long force pulse of 600 nN applied to CIPS/Ti resulted in the clear formation of oppositely polarized domains, driven by the inhomogeneous nucleation of new downward-polarized regions, thus confirming mechanical switching. This switching behavior was further validated by additional measurements (Supplementary Fig. 2). Increasing the force pulse to 1000 nN enabled switching of >95% of the area to downward polarization (Supplementary Fig. 3). In contrast, a 1000 nN force pulse applied to CIPS/Au, which lacks the imprint field, barely induced polarization switching, with only a few small downward-polarized domains observed. Notably, the CIPS topography remained intact after continuous application of multiple 600 nN pulses, whereas the surface sustained damage after several 1000 nN force pulses (Supplementary Fig. 4).

Recent experiment reveals that the threshold force for flexoelectric-induced polarization switching increases linearly with ferroelectric thickness in the ultrathin regime[50]. In thicker ferroelectrics, the threshold force tends to saturate with further increases in thickness. Therefore, normalizing the threshold force by the film thickness, i.e., force per unit thickness, provides a simple and effective metric for benchmarking threshold forces across ferroelectric systems with varying thicknesses. The low-force switching achieved in this work is therefore highlighted by comparing the force per unit thickness required for polarization switching in reported low-dimensional

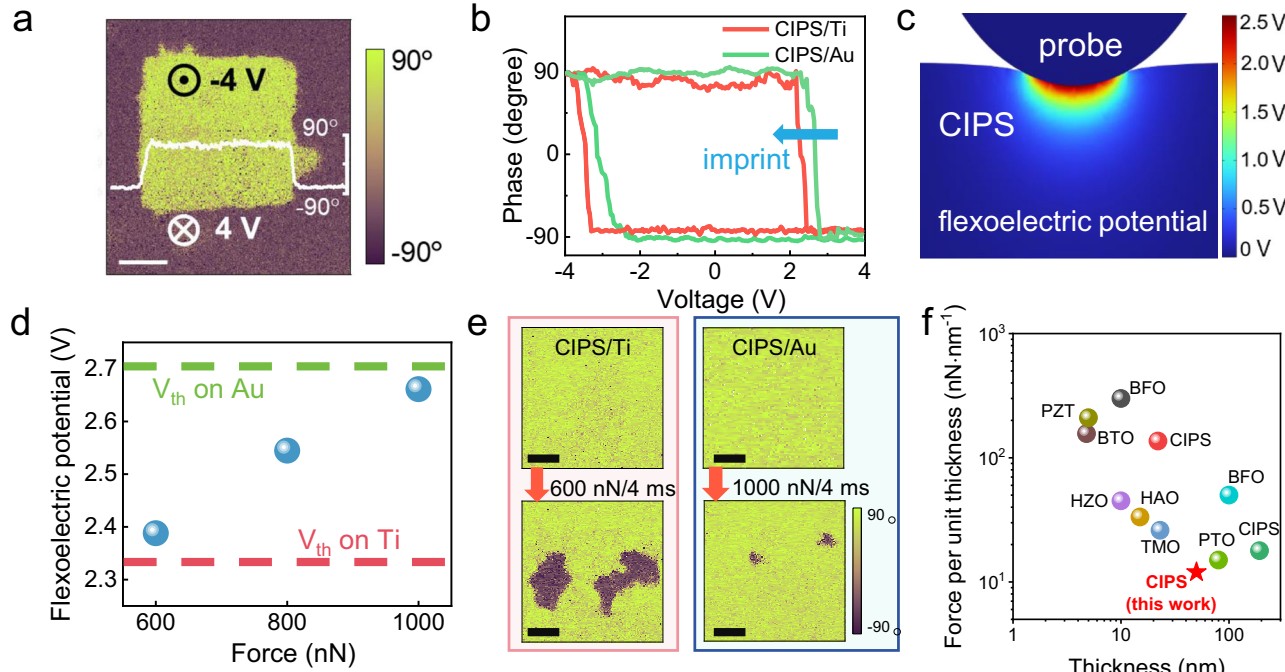

**Fig. 2 | Low-force mechanical switching of polarization. a** Piezoresponse force microscopy (PFM) phase image showing an oppositely polarized pattern written by electrical bias voltages of ±4 V. Scale bar: 1 μm. The inset curve shows the phase profile along a line scan. **b** Local phase hysteresis loops of CIPS measured on Ti and Au. Clear imprint effect is observed for CIPS/Ti sample. **c** Finite element simulation calculated vertical flexoelectric potential distribution in the deformed CIPS under a 600 nN force application. **d** Flexoelectric potential generated at the surface of CIPS subjected to various applied forces. Dashed lines indicate the threshold voltages required for polarization switching on Ti and Au. **e** PFM phase images of CIPS on Ti and Au, measured before (upper) and after (lower) the 600 nN and 1000 nN force pulses. Scale Bar: 200 nm. **f** Comparison of the force per unit thickness required for mechanical polarization switching in various thin ferroelectrics.

ferroelectrics, as illustrated in Fig. 2f[4,6,30,32,33,41,42,51–53]. Note that, for studies involving thicker ferroelectrics that enter the saturation regime, the force per unit thickness represents an underestimate of the actual level of threshold force. Our work achieves a record-low threshold force per thickness of 12 nN·nm$^{-1}$, marking a significant improvement over previous studies, which is attributed to the synergistic combination of utilizing two-dimensional ferroelectric CIPS and an asymmetric boundary design. The low threshold force per unit thickness for mechanical switching was further verified using a thinner 24 nm-thick CIPS sample (Supplementary Fig. 5). A single 4 ms-long force pulse of 300 nN was sufficient to induce mechanical polarization switching (Supplementary Fig. 6), corresponding to a threshold force per unit thickness of 12.5 nN·nm$^{-1}$, which is in good agreement with the ~12 nN·nm$^{-1}$ value obtained from the 50 nm-thick sample. These results provide additional experimental evidence supporting the linear scaling behavior of the threshold force in ultrathin ferroelectrics. More importantly, compared to previous studies, which typically require the static force applications in the range of tens to thousands of milliseconds to achieve polarization switching, our approach demonstrates a remarkable improvement in speed and efficiency. Such improvements not only mitigate material degradation risks associated with high-force applications but also enhance the feasibility of integrating flexoelectric effects into functional devices requiring rapid and precise control of polarization states.

The ability to gradually control polarization switching with fast, low-force pulses is a critical feature for realizing multiple conductance states in emerging ferroelectric device applications. To further investigate this, a series of 4 ms-long force pulses of 600 nN were applied to the surface of CIPS/Ti, initially set in the up-polarized state by a negative voltage bias. By analyzing PFM phase images acquired after different cumulative pulse durations, the gradual switching of ferroelectric domains was observed, as shown in Fig. 3a. A nearly linear correlation was found between the mechanically switched domain

area and the cumulative pulse duration. Specifically, one 4 ms-long pulse switched ~20% of the up-polarized domains to the downward direction, with the reversed domain area increasing to ~80% after a cumulative pulse duration of 48 ms (12 applied pulses). In contrast, CIPS on Au remained in the up-polarized state after multiple pulses, as the force pulse was insufficient to overcome the switching energy barrier (Supplementary Fig. 7). The spatial control of force application enables the writing of designated patterns with down-polarized regions. As shown in Fig. 3b, PFM phase images reveal the down-polarized "CSU" letters written on the CIPS surface by applying a force pulse train to defined regions. Furthermore, electrical voltage can be applied to switch the polarization back to the upward direction, effectively erasing the information written by mechanical force. Figure 3c demonstrates the PFM phase images of CIPS subjected to one cycle of "mechanical-writing" and "electrical-erasing" operations, verifying the reversible and programmable nature of this approach.

**Mechanically gated multi-level ferroelectric transistor**

Finally, we demonstrate the application of the asymmetric contact configuration in a semiconductor-ferroelectric heterostructure to prototype a mechanically gated ferroelectric transistor controlled by low force pulses. A van der Waals heterostructure was constructed by consecutively stacking graphene, few-layer MoS$_2$, and CIPS on a SiO$_2$ substrate, where graphene, MoS$_2$, and CIPS serve as the electrical contacts, semiconducting channel, and ferroelectric layer, respectively, analogous to the structure of a conventional ferroelectric transistor[54–56]. The optical microscope image of a fabricated device is presented in Fig. 4a. Unlike conventional ferroelectric transistors that rely on electrical gating, this device employs a Pt probe as a mechanical gate to modulate the channel conductance through applied mechanical force. The energy alignment between Pt, CIPS, and MoS$_2$ is critical to the device functionality, as shown in Fig. 4a. Few-layer MoS$_2$, with a Fermi energy of 4.4–4.5 eV[57,58], aligns slightly below the work function

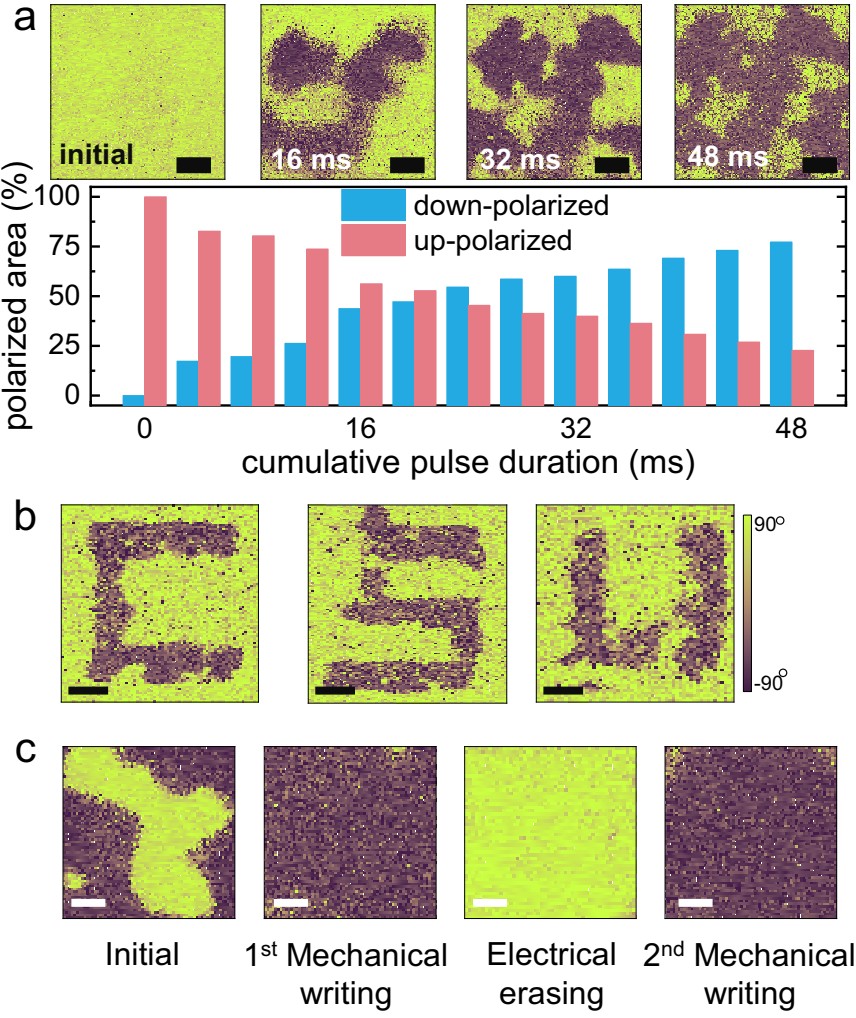

**Fig. 3 | Low force pulse-controlled polarization switching. a** Force pulses-controlled partial polarization switching in CIPS. PFM phase images of CIPS subjected to multiple 4 ms-long force pulses of 600 nN. Scale bar: 200 nm. The variations of the areas of up- and down-polarization are extracted from the PFM images for specific cumulative pulse durations. **b** PFM phase images showing the "CSU" patterns written by force pulses at the designated areas indicated by the dashed boxes. Scale bar: 200 nm. **c** PFM phase images of CIPS demonstrating a cycle of mechanical writing – electrical erasing operation. Scale bar: 200 nm.

of Ti. Therefore, an imprint field is created in the CIPS layer that facilitates low-force downward polarization switching.

The mechanical gate tunability is experimentally varied in the transistor presented in Fig. 4a with a spontaneously polarized CIPS layer (Fig. 4b). From the PFM phase image, an up-polarized domain is initially observed at the center of the CIPS, spanning the entire width of the channel. The Fermi level of $MoS_2$ is pushed toward the valence band under the up-polarized region, which blocks the carrier transport in $MoS_2$, resulting in a high-resistance state with a conductance of 0.035 nS. Then, a 100 ms-long low-force pulse is then applied at mechanical gate, which effectively switches the area into downward polarization, as evidenced in the PFM phase image in Fig. 4b. Subsequently, the Fermi level is shifted upward in $MoS_2$ by the downward polarization. Hence, carrier transport is facilitated in the $MoS_2$, increasing the channel conductance to 0.32 nS, corresponding to an on/off ratio of ~10.

A key feature of this device is its ability to achieve nonvolatile, multi-level conductance states through partial polarization switching under controlled force pulses. The optical microscope image of the measured device is presented in Supplementary Fig. 8. The CIPS layer was initialized to an upward polarization state prior to the application of mechanical force pulses, thereby setting the $MoS_2$ channel in a high-resistance state. Subsequently, three sequential force pulses were applied via mechanical gating. Each pulse gradually switches the CIPS polarization toward the downward direction as illustrated in Fig. 3a. The regions of $MoS_2$ beneath the down-polarized CIPS domains exhibit a low-resistance state. Accordingly, increasing the number of applied force pulses progressively expands the down-polarized domains in the CIPS layer, resulting in a corresponding increase in the high-conductance regions within the $MoS_2$ channel. As a result, by applying three sequential force pulses, we demonstrate four distinct conductance levels, corresponding to a 2-bit data stream from "00" to "11" (Fig. 4c). Similar multi-level conductance states have been achieved in conventional ferroelectric field-effect transistors, where gradual polarization switching was controlled through the application of gate voltage pulses[56]. These conductance states exhibit excellent non-volatility, maintaining stability for over 1000 s, which is critical for applications in tactile-sensitive neuromorphic hardware and multi-bit data storage. An additional device was fabricated and measured to verify the reproducibility of the multi-level conductance states controlled by applied force pulses (Supplementary Fig. 9). Three distinct, nonvolatile conductance states were demonstrated by gradually switching the polarization of CIPS through sequential force pulses (Supplementary Fig. 10).

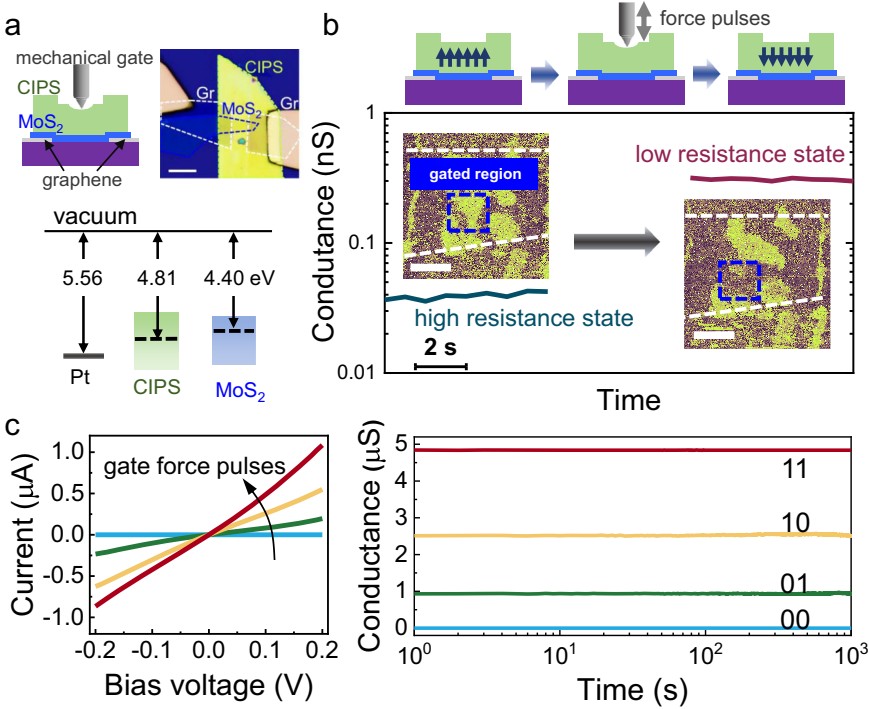

**Fig. 4 | A mechanical gated transistor with multi-level conductance.**
**a** Schematic of a mechanically gated transistor comprising a $MoS_2$ channel, a CIPS ferroelectric layer, and a Pt-probe mechanical gate. Optical microscope image of a fabricated transistor. The dashed lines provide the eye guides to the edges of each functional layers. Scale bar: 5 μm. Energy band diagrams show the alignment of energy levels in Pt, CIPS, and $MoS_2$ before and after contact. **b** Conductance of the transistor measured before and after gate force operation, showing high- and low-resistance states. Insets display corresponding PFM phase images of the CIPS on top of the $MoS_2$ channel, with thick dashed lines indicating the edges of the underlying $MoS_2$ channel along the current flow path. The gated region is highlighted by the blue dashed box. Clear polarization switching is observed due to gate force application. Scale bar: 1 μm. **c** Output curves of the transistor controlled by gate pulse operations, exhibiting four distinct conductance states. Retention test results of the four conductance states, corresponding to 2 bit data, measured up to 1000 s.

## Discussion

In summary, we have developed an effective strategy to lower the threshold force for flexoelectricity-induced polarization switching in CIPS by introducing an imprint field through an asymmetric contact configuration. Using 4 ms low-force pulses of 600 nN, we successfully achieved polarization switching in 50 nm-thick CIPS between Pt and Ti without causing material damage, facilitated by the imprint field. In contrast, polarization switching could not be achieved in CIPS with symmetric contacts, even under strong force pulses of >1000 nN, which exceed the mechanical tolerance of CIPS. A much-lowered threshold force per thickness of 12 nN·nm$^{-1}$, combined with the ability to perform polarization switching using rapid 4 ms force pulses, highlights the groundbreaking potential of enabling energy-efficient and high-speed mechanical control of ferroelectric devices. The imprint field concept is further extended to a CIPS/$MoS_2$ heterostructure, enabling the development of a mechanically gated ferroelectric transistor, which demonstrates nonvolatile, multi-level conductance states controlled by low-force mechanical pulses. These results highlight the potential of integrating mechanical tunability and tactile functionality into ferroelectric transistors, paving the way for next-generation smart systems with enhanced energy efficiency and tactile sensitivity.

## Methods

### CIPS/metal samples preparation

The metal substrates were prepared by evaporating gold and titanium films on the silicon substrates. Then CIPS samples were mechanically exfoliated from a commercially sourced bulk single crystal (Six Carbon Technology, Shenzhen) and transferred onto target substrates. The use of single crystalline samples eliminates undesired effects caused by grain boundaries. The 50 nm-thick CIPS flakes were selected using an optical microscope. The thickness was further confirmed by atomic force microscopy (see Supplementary Fig. 11). Here, 50 nm-thick CIPS samples were employed, as they were more easily obtained by mechanical exfoliation compared to thinner flakes, while offering good reproducibility and negligible thickness variation.

### Scanning probe measurements

All the scanning probe measurements, including AFM, KPFM, and PFM, were conducted using a Nanosurf CoreAFM system in the corresponding modes. Specifically, PFM measurements were carried out with the overall platinum-coated Multi75E-G tip with a spring constant of 3 N·m$^{-1}$ under an AC voltage of 1.5 V, that is below the coercive voltage of CIPS. A low tip force of 20 nN was applied to the surface of CIPS during the PFM measurements to prevent unintended flexoelectric-induced switching.

### Mechanically gated transistor fabrication and characterization

Fabrication begins with the preparation of graphene contacts. Graphene flakes were mechanically exfoliated from single-crystal HOPG (HQ Graphene Groningen) and transferred onto a $SiO_2$/Si substrate. Two adjacent few-layer graphene flakes, defined by plasma etching, served as van der Waals contacts. A few-layer $MoS_2$ flake was exfoliated from the single crystal (HQ Graphene Groningen) and then was aligned under an optical microscope and transferred onto the graphene to function as the semiconducting channel. Next, a 50 nm-thick CIPS flake was stacked on top of the $MoS_2$ as the gate dielectric. Finally, metal electrodes were defined on graphene contacts with a 5 nm/30 nm chrome/gold stack by photolithography and evaporation. The fabricated devices were wire bonded on a printed circuit board. The

electrical measurements were carried out in the Nanosurf scanning probe system under ambient conditions using a source-measure unit. Mechanical gate force was applied to the surface of CIPS by scanning the PFM tip over a square region, using a specified setpoint force along with designated scan steps and speed.

**Finite element simulation and flexoelectric potential calculation**
Finite element simulations were carried out using the solid mechanics module of COMSOL software. A contact model consisting of a hemisphere tip with the radius of 25 nm and a 50 nm-thick CIPS sample was constructed to study the flexoelectric potential generated by the tip force. The force was applied by adding a static force load on the tip. The material parameters, i.e., Young's modulus $E$, Poisson's ratio $v$, and density $\rho$, are obtained from previous reports and listed as follows[38,41,59]. For CIPS, $E_c = 29$ GPa, $v_c = -0.06$, and $\rho_c = 3429$ kg·m$^{-3}$. For tip, $E_t = 170$ GPa, $v_t = 0.3$, and $\rho_t = 2200$ kg·m$^{-3}$.

The flexoelectric potential can be calculated using the following equation[3]:

$$V_{\text{flexo}} = \frac{f_{3311}e_{xx} + f_{3322}e_{yy} + f_{3333}e_{zz}}{\varepsilon_0 \varepsilon_\gamma}$$

where $e_{xx}$, $e_{yy}$, $e_{zz}$ are the strain distributions along various directions caused by the tip-induced deformation, which are obtained from the finite element simulations. The detailed derivation of the flexoelectric voltage can be found in Supplementary Note 1. Supplementary Fig. 12 plots the strain distributions under the force load of 600 nN. Dielectric constant $\varepsilon_\gamma$ is 40 for CIPS, and $\varepsilon_0$ is the permittivity of vacuum. Flexoelectric coefficients have been reported to lie in the range of 1 nC·m$^{-1}$ to 10 nC·m$^{-1}$ in previous studies[41,59]. In the simulations, their values are set as $f_{3311} = 6$ nC·m$^{-1}$, $f_{3322} = 2$ nC·m$^{-1}$, and $f_{3333} = 2$ nC·m$^{-1}$, in order to achieve the best agreement with the experimental results.

## Data availability
All the data supporting the findings of this study are available within the article and its Supplementary Information. The data generated in this study are provided in a Source Data file. Source data are provided with this paper.

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

## Acknowledgements

This work was supported by National Key R&D Program of China (Grant No. 2022YFA1405600), National Natural Science Foundation of China (Grant No. 12374051), Hunan Provincial Science and Technology Department (Grant Nos. 2023JJ40773, 2023JJ20070) and Natural Science Foundation of Changsha (Grant No. kq2208254).

## Author contributions

J.S. conceived the project. Y.Z. carried out the measurements and analyzed the data. S.D., X.L., Y.Y., Y.J., T.T., F.X., K.C. provided support for measurements and data analyses. Y.Z. and J.S. wrote the manuscript with contributions from T.T., J.C., and F.X. All authors discussed the results and contributed to the manuscript.

## Competing interests

The authors declare no competing interests.
