## [Transparent Peer Review file · Nature Communications]

Low-Force Pulse Switching of Ferroelectric Polarization Enabled by Imprint Field

Corresponding Author: Professor Jian Sun

Version 0:

Reviewer comments:

Reviewer #1

(Remarks to the Author)

The manuscript “Low-Force Pulse Switching of Ferroelectric Polarization Enabled by Imprint Field” reports on experimentally confirmed local ferroelectric switching by indentation. The study stands out by the minimal force per thickness of the material needed for switching. This is achieved via introducing a bias through asymmetric top and bottom electrodes with different workfunctions. The reverse switching or “erasing” is however performed electrically.

The findings are solid and the presentation is generally good. There are however few drawbacks which should be corrected.

1 The authors state that force per thickness of the material is an important indicator but do not explain why. Scaling laws of the kind indeed are known for flexoelectricity, for example in [10.1016/j.ijengsci.2022.103771] it is shown that deflection of a membrane due to converse flexoelectric effect is inverse proportional to the square of its thickness. This follows from equations. Could the authors provide a simple equation for their scaling law? Or at least provide a reference where such a law is derived? Otherwise the major statement of the authors, summarized in Fig. 2f is not valid.

2 The authors should remove statement that the work “advances the understanding of flexoelectricity...” and make it clear that their theoretical part is a tentative reasonable simplistic interpretation. The formula that the authors use was taken from supplementary materials of Ref. 3 apparently without any critical assessment. Could the authors start from a commonly accepted grounds and justify the applicability of their approximation? Generally, a potential may be introduced only to a curlless field which is not the case for a common flexoelectric field. Some assumptions were clearly made, which ones? The authors apparently introduced the values for flexoelectric moduli in their material to best fit the experimental results. It would be wise to state directly that the experimental results are fit with the values for flexoelectric coefficients which are in good agreement with those known from literature (6 nC/m versus 10 nC/m) as compared to Ref [41].

3 There is a typo: “Dielectric constant ϵ_0 is 40 for CIPS”, must be ϵ_y .

After the aforementioned drawbacks are corrected the article is recommended for publication.

Reviewer #2

(Remarks to the Author)

The authors propose a strategy to reduce the force needed for flexoelectric polarization switching using an asymmetric boundary design that generates a controlled imprint field, stabilizing the energy landscape and lowering the switching barrier. Their approach enables effective polarization switching in CIPS with low-force pulses (600 nN) and a record low threshold force of 12 nN/nm. Without the imprint field, higher forces (>20 nN/nm) fail to switch polarization and cause damage. The design is compatible with electronic devices. Also, they demonstrate a mechanically-gated ferroelectric transistor with multi-level conductance controlled by low-force stimuli, advancing the field of mechanically controlled devices. Some good advancements in the field are realized in this manuscript which I believe is truly worth considering for the potential publication however provided the following comments should be considered:

1. A proper description of the process which includes mechanical gate force application via the probe is missing. The authors should clarify it.

2. Why a 50 nm-thick CIPS flake was used as a gate dielectric? Is there any thickness-related hypothesis behind this selection? Did the authors try other thicknesses for the fabrication and what was the outcome?
3. How does the application of three sequential force pulses to a transistor result in different conductance states? The physics behind seems poorly demonstrated.
4. How/does the imprint field change by the metal type and CIPS flak dimensions?
5. How does the grain boundary of CIPS affect the switching properties? Authors should briefly describe it in the manuscript.

Reviewer #3

(Remarks to the Author)

The work by Zhang et al. presents the case of CuInP_2S_6 , a two-dimensional (2D) ferroelectric material with spontaneous out-of-plane electric polarization, and its potential use as a flexoelectric material. Their results report the lowest threshold force per thickness by the implementation of a strategy based on rapid, low force pulses to control of the imprint field via asymmetric boundary design with little damage to the sample. They also show the potential applications of the technique in functional gated devices based on van der Waals heterostructures (made purely using stacked 2D materials), where the switch in polarization in the ferroelectric material is induced by the mechanical strain and not by an external electric field.

The manuscript is well written, easy to understand and provides an extensive literature review. As the authors deal with a 2D ferroelectric material, I believe this work can appeal to a broader audience in the field of two-dimensional materials and devices. However, because of this appeal to the condensed matter community in general, I would like the authors to address my concerns before publication:

1. It is clear that the threshold force is a relevant factor when dealing with ferroelectric materials, and trying to minimize it would benefit their implementation in commercial devices. However, I miss a proper definition of what an imprint field is and how it is relevant (it is not very clear to me in the text).
2. I found in the literature that having zero imprint field benefits the functionality of ferroelectric-based devices [Sci. Rep. 6, 25028 (2016)], but the authors claim that this imprint field is important for allowing a relatively easy polarization switching. I find both statements contradictory, so I would like the authors to comment on this and clarify it.
3. I find appealing the application of the described technique using van der Waals heterostructures, as it is known that applying high electric fields in CIPS can induce the migration. Did the authors just measure one sample? From the text, I assume only one device was prepared and tested (as shown in Figure S7). When dealing with devices, it is very important to test the reproducibility of the observed phenomena in more than one sample, so I strongly recommend to at least measure another with the same configuration proving that the several samples show the same gate modulation when applying the force pulses. I would also like to see the image of the device as part of Figure 4, as I think it provides a visual representation of the experiment.
4. I think it is important to specify the source of the CIPS crystals (can be written in the methods section); were they synthesized by the authors, or were they obtained from a company? Same goes for the MoS_2 and HOPG.
5. Please, write the units of the colour scale in Figure S8.
6. Just a curiosity: did the authors choose CIPS instead of other layered ferroelectric materials (i.e. In_2Se_3) for any particular reason?

If the authors address my concerns (specially point 3, which I believe is the most exciting part of the work if developed further), I can recommend the publication of the manuscript in Nature Communications. If it is not possible to prove the claim regarding the devices providing more samples, I would suggest to transfer the article to npj Flexible Electronics as I think it fits better with the 'Aims & Scope' of the journal.

Version 1:

Reviewer comments:

Reviewer #1

(Remarks to the Author)

The authors have taken into account all my criticisms. Remarks from the other referees are also seemingly treated correctly. The article is now recommended for publication.

Reviewer #2

(Remarks to the Author)

In my opinion authors addressed my concerns acceptably. Thus, I recommend this work for the publication.

Reviewer #3

(Remarks to the Author)

The authors have addressed each one of my concerns and they have provided detailed answers to my questions. Therefore, I can recommend the publication of the article with the provided corrections in Nature Communications.

RESPONSES TO REVIEWERS' COMMENTS:

Reviewer #1 (Remarks to the Author):

The manuscript “Low-Force Pulse Switching of Ferroelectric Polarization Enabled by Imprint Field” reports on experimentally confirmed local ferroelectric switching by indentation. The study stands out by the minimal force per thickness of the material needed for switching. This is achieved via introducing a bias through asymmetric top and bottom electrodes with different work functions. The reverse switching or “erasing” is however performed electrically. The findings are solid and the presentation is generally good. There are however few drawbacks which should be corrected.

The authors state that force per thickness of the material is an important indicator but do not explain why. Scaling laws of the kind indeed are known for flexoelectricity, for example in [10.1016/j.ijengsci.2022.103771] it is shown that deflection of a membrane due to converse flexoelectric effect is inverse proportional to the square of its thickness. This follows from equations. Could the authors provide a simple equation for their scaling law? Or at least provide a reference where such a law is derived? Otherwise the major statement of the authors, summarized in Fig. 2f is not valid.

Answer: Thank you for bringing in this important issue. The mechanical strain gradient generated by surface force application through a probe tip decreases rapidly with depth in ferroelectric materials. As a result, the threshold force required for polarization switching is dependent on the film thickness. We attempted to normalize the threshold force with respect to film thickness, given that ferroelectric films used in previous studies span a wide range—from 10 nm to 100 nm. However, the scaling law of the threshold force remains incomplete and is not yet fully understood in the context of the flexoelectric effect [Acta Materialia 193, 151-162 (2020); Appl. Phys. Rev. 8, 041327 (2021)]

As known that flexoelectrical field is expressed as

$$E_{\text{flexo}} = \frac{f_{3311}}{\epsilon_0 \epsilon_\gamma} \frac{\partial e_{xx}}{\partial z} + \frac{f_{3322}}{\epsilon_0 \epsilon_\gamma} \frac{\partial e_{yy}}{\partial z} + \frac{f_{3333}}{\epsilon_0 \epsilon_\gamma} \frac{\partial e_{zz}}{\partial z}.$$

Here, f is the flexoelectric tensor, $\epsilon_0 \epsilon_\gamma$ is dielectric constant, e is the strain. In the simplified case of a uniform force F applied to a solid, the resulting strain is proportional to F . However, to induce a strain gradient, which is essential for flexoelectric effect, a nonuniform force is

necessary, such as one applied via a probe tip. Consequently, a simple scaling law for the flexoelectric effect is difficult to derive. According to experimental investigations by Long-Qing Chen *et. al*, the flexoelectric field exhibits a nontrivial dependence on the applied force using a probe tip across different film thicknesses [Acta Materialia 193, 151-162 (2020)]. For ultrathin films, the threshold force increases approximately linearly with thickness. However, in thicker films, this threshold force either saturates or even decreases, which is counterintuitive. The underlying mechanisms remain unclear but may involve misfit strain relaxation, polarization relaxation near surfaces, or the emergence of alternative mechanisms. Nevertheless, we chose to normalize the threshold force by the film thickness, under the assumption that the thicknesses reported in previous studies fall within the linear-response regime. In cases where the threshold force in thicker films (e.g., these ones thicker than 100 nm) has already entered the saturation regime, the normalized force per unit thickness, as plotted in Figure 2f, may represent an underestimate.

Figure R1. AFM images of a thinner CIPS sample on Ti substrate, the line height profile indicates the thickness of ~24 nm.

To verify that the thickness dependence of the threshold force lies within the linear regime in our work, we conducted experiments on a thinner CIPS sample with a thickness of 24 nm (See Figure R1). Figure R2 illustrate the PFM phase images of CIPS/Ti obtained after initialization and after sequential force applications of 300 nN and 400 nN of 4 ms. As shown, a single 4 ms-long force pulse of 300 nN applied to the CIPS/Ti sample resulted in the clear formation of oppositely polarized domains, covering 18.7% of the area. This change confirms mechanical polarization switching. In contrast, a weaker force of 200 nN was insufficient to

induce switching. A stronger force of 400 nN further increased the switched domain area to 56.2%, indicating enhanced polarization reversal. These results suggest that the threshold force lies slightly below 300 nN. Accordingly, the threshold force per unit thickness for the 24 nm-thick CIPS sample is estimated to be slightly below ~ 12.5 nN/nm, which is in good agreement with the value of ~ 12 nN/nm obtained from the 50 nm-thick sample. Taken together with the findings from Chen et al., our results support the use of force per unit thickness as a meaningful normalization parameter for evaluating the threshold force required for flexoelectric polarization switching.

Figure R2. PFM phase images of the 24 nm-thick CIPS on Ti obtained after initialization and after sequential force applications of 4 ms-long 300 nN and 400 nN. Size: $1\ \mu\text{m} \times 1\ \mu\text{m}$. In particular, a single 4 ms-long force pulse of 300 nN results in the formation of oppositely polarized domains, covering 18.7% of the area, confirming mechanical polarization switching. A stronger force of 400 nN further increases the switched domain area to 56.2%.

We have added the following discussion in the revised manuscript regarding the normalization. Figure R1 and Figure R2 have been added in the revised Supplementary Information as Fig. S5 and Fig. S6.

“Recent experiment reveals that the threshold force for flexoelectric-induced polarization switching increases linearly with ferroelectric thickness in the ultrathin regime [50]. In thicker ferroelectrics, the threshold force tends to saturate with further increases in thickness. Therefore, normalizing the threshold force by the film thickness, i.e., force per unit thickness, provides a simple and effective metric for benchmarking threshold forces across ferroelectric systems with varying thicknesses.”

“Note that, for studies involving thicker ferroelectrics that enter the saturation regime, the force per unit thickness represents an underestimate of the actual level of threshold force.”

“The low threshold force per unit thickness for mechanical switching was further verified using a thinner 24 nm-thick CIPS sample (Fig. S5). A single 4 ms-long force pulse of 300 nN was sufficient to induce mechanical polarization switching (Fig. S6), corresponding to a threshold force per unit thickness of 12.5 nN/nm, which is in good agreement with the ~12 nN/nm value obtained from the 50 nm-thick sample. These results provide additional experimental evidence supporting the linear scaling behavior of the threshold force in ultrathin ferroelectrics.”

The authors should remove statement that the work “advances the understanding of flexoelectricity” and make it clear that their theoretical part is a tentative reasonable simplistic interpretation. The formula that the authors use was taken from supplementary materials of Ref. 3 apparently without any critical assessment. Could the authors start from a commonly accepted grounds and justify the applicability of their approximation? Generally, a potential may be introduced only to a curlless field which is not the case for a common flexoelectric field. Some assumptions were clearly made, which ones? The authors apparently introduced the values for flexoelectric moduli in their material to best fit the experimental results. It would be wise to state directly that the experimental results are fit with the values for flexoelectric coefficients which are in good agreement with those known from literature (6 nC/m versus 10 nC/m) as compared to Ref [41].

Answer: Thank you for your suggestion. We have revised the statement to avoid overestimating the theoretical contribution of this work in the revised manuscript as below.

“This work offers a pathway for effectively controlling polarization in low-dimensional ferroelectrics via the flexoelectric effect and provides a foundation for developing mechanically controlled ferroelectric devices.”

We are sorry for the missing details of the theoretical model. Here we start from the general framework of flexoelectricity to derive the flexoelectric voltage equation presented in the Methods section. Flexoelectricity depends on gradients of strain, the component of the

induced polarization can be expressed as $P_i = f_{ijkl} \frac{\partial e_{kl}}{\partial x_j}$, where f_{ijkl} is flexoelectric coefficient tensor, e_{kl} is the strain tensor, x_j is the coordinate. Hence, the bound charge density is expressed as $\rho = -\nabla \cdot \mathbf{P}$. The resulting flexoelectric field \mathbf{E} is therefore calculated through Gauss's law as $\mathbf{E} = -\frac{\mathbf{P}}{\epsilon_0 \epsilon_\gamma}$ with $\epsilon_0 \epsilon_\gamma$ being the dielectric constant of the ferroelectric. The flexoelectric voltage is then the line integral of the field, $V = -\int \mathbf{E} \cdot d\mathbf{l}$. In the case of probe tip induced force on a thin film with out-of-plane polarization, only out-of-plane polarization P_z is considered and strain gradients are dominated by the out-of-plane direction z . Hence, all the lateral field component can be neglected. Flexoelectric coupling reduces to effective constants f_{33jj} , *i.e.*, flexoelectric coefficient in z -direction from various strain components. Then polarization can be simplified as $P_z = f_{3311} \frac{\partial e_{xx}}{\partial z} + f_{3322} \frac{\partial e_{yy}}{\partial z} + f_{3333} \frac{\partial e_{zz}}{\partial z}$. Subsequently, the flexoelectric field becomes $E_{\text{flexo}} = \frac{f_{3311}}{\epsilon_0 \epsilon_\gamma} \frac{\partial e_{xx}}{\partial z} + \frac{f_{3322}}{\epsilon_0 \epsilon_\gamma} \frac{\partial e_{yy}}{\partial z} + \frac{f_{3333}}{\epsilon_0 \epsilon_\gamma} \frac{\partial e_{zz}}{\partial z}$. And finally, the flexoelectric voltage is expressed as $V_{\text{flexo}} = \frac{f_{3311} e_{xx} + f_{3322} e_{yy} + f_{3333} e_{zz}}{\epsilon_0 \epsilon_\gamma}$.

In this analysis, it also assumes quasi-static conditions and a spatially homogeneous and isotropic medium, where the strain gradient-induced polarization varies smoothly in space. Under these assumptions, the induced flexoelectric field remains conservative and hence curl-free, allowing the use of a scalar potential. This simplification is commonly employed in flexoelectric literature to facilitate analytical modeling.

We have added the following statement regarding the theoretical model in the revised SI as SI Note 1.

“The detailed derivation of the flexoelectric voltage can be found in Supplementary Note 1.”

“Flexoelectricity depends on gradients of strain, the component of the induced polarization can be expressed as $P_i = f_{ijkl} \frac{\partial e_{kl}}{\partial x_j}$, where f_{ijkl} is flexoelectric coefficient tensor, e_{kl} is the strain tensor, x_j is the coordinate. Hence, the bound charge density is expressed as $\rho = -\nabla \cdot \mathbf{P}$. The resulting flexoelectric field \mathbf{E} is therefore calculated through Gauss's law as $\mathbf{E} = -\frac{\mathbf{P}}{\epsilon_0 \epsilon_\gamma}$ with $\epsilon_0 \epsilon_\gamma$ being the dielectric constant of the ferroelectric. The flexoelectric voltage is then the line integral of the field, $V = -\int \mathbf{E} \cdot d\mathbf{l}$. In the case of probe tip induced

force on a thin film with out-of-plane polarization, only out-of-plane polarization P_z is considered and strain gradients are dominated by the out-of-plane direction z . Hence, all the lateral field component can be neglected. Flexoelectric coupling reduces to effective constants f_{33jj} , *i.e.*, flexoelectric coefficient in z -direction from various strain components. In this analysis, it also assumes quasi-static conditions and a spatially homogeneous and isotropic medium, where the strain gradient-induced polarization varies smoothly in space. Under these assumptions, the induced flexoelectric field remains conservative and hence curl-free, allowing the use of a scalar potential. Then polarization can be simplified as

$P_z = f_{3311} \frac{\partial e_{xx}}{\partial z} + f_{3322} \frac{\partial e_{yy}}{\partial z} + f_{3333} \frac{\partial e_{zz}}{\partial z}$. Subsequently, the flexoelectric field becomes

$E_{\text{flexo}} = \frac{f_{3311}}{\epsilon_0 \epsilon_\gamma} \frac{\partial e_{xx}}{\partial z} + \frac{f_{3322}}{\epsilon_0 \epsilon_\gamma} \frac{\partial e_{yy}}{\partial z} + \frac{f_{3333}}{\epsilon_0 \epsilon_\gamma} \frac{\partial e_{zz}}{\partial z}$. And finally, the flexoelectric voltage is expressed as

$$V_{\text{flexo}} = \frac{f_{3311} e_{xx} + f_{3322} e_{yy} + f_{3333} e_{zz}}{\epsilon_0 \epsilon_\gamma} .$$

Compared to conventional ferroelectric oxides, the material parameters of CIPS have not been thoroughly investigated, particularly with respect to the flexoelectric effect. Some of the material parameters used in the finite element simulations, such as Young's modulus, Poisson's ratio, density, and dielectric constant, were set to the certain values reported in the literature [Sci. Adv. 8, eabq1232(2022); Adv. Mater. 35, 2302320 (2023); Nat. Commun. 13, 574 (2022)]. However, uncertainty remains regarding the flexoelectric coefficient components. Previous studies reported values in the range of 1~10 nC/m. In this work, the coefficients were selected within this range, with specific values chosen to achieve the best agreement with experimental results.

We have added the following discussion in the revised manuscript:

“The flexoelectric coefficients have been reported to lie in the range of 1 nC/m to 10 nC/m in previous studies [41,59]. In the simulations, their values are set to $f_{3311} = 6$ nC/m, $f_{3322} = 2$ nC/m, and $f_{3333} = 2$ nC/m, in order to achieve the best fit with the experimental results.”

There is a typo: “Dielectric constant ϵ_0 is 40 for CIPS” , must be ϵ_γ .

Answer: We are sorry for the typo. In the revised manuscript, it has been corrected to

“Dielectric constant ϵ_r is 40 for CIPS”.

Reviewer #2 (Remarks to the Author):

The authors propose a strategy to reduce the force needed for flexoelectric polarization switching using an asymmetric boundary design that generates a controlled imprint field, stabilizing the energy landscape and lowering the switching barrier. Their approach enables effective polarization switching in CIPS with low-force pulses (600 nN) and a record low threshold force of 12 nN/nm. Without the imprint field, higher forces (>20 nN/nm) fail to switch polarization and cause damage. The design is compatible with electronic devices. Also, they demonstrate a mechanically-gated ferroelectric transistor with multi-level conductance controlled by low-force stimuli, advancing the field of mechanically controlled devices. Some good advancements in the field are realized in this manuscript which I believe is truly worth considering for the potential publication however provided the following comments should be considered:

1. A proper description of the process which includes mechanical gate force application via the probe is missing. The authors should clarify it.

Answer: Thank you for this comment. We apologize for the lack of detail regarding the mechanical gate force application. In our experiments, the force was applied to the surface of CIPS using a Pt-coated Multi75E-G PFM probe with a spring constant of 3 N/m. Mechanical gating was performed by scanning the tip over a defined square region under a specified setpoint force. To prevent unintended flexoelectric-induced switching during PFM imaging, the tip force was significantly reduced to 20 nN for domain visualization. The duration of force application was controlled by adjusting the scan step size and speed.

We have added a more detailed description of the mechanical gate force application in the revised Methods section.

“Specifically, PFM measurements were carried out with the overall platinum-coated Multi75E-G tip with a spring constant of 3 N/m under an AC voltage of 1.5 V, that is below the coercive voltage of CIPS. A low tip force of 20 nN was applied to the surface of CIPS during the PFM measurements to prevent unintended flexoelectric-induced switching.”

“The electrical measurements were carried out in the Nanosurf scanning probe system under ambient conditions using a source-measure unit. Mechanical gate force was applied to the surface of CIPS by scanning the PFM tip over a square region using a specified setpoint force. The duration of the force application was controlled by adjusting the scan step size and scanning speed.”

2. Why a 50 nm-thick CIPS flake was used as a gate dielectric? Is there any thickness-related hypothesis behind this selection? Did the authors try other thicknesses for the fabrication and what was the outcome?

Answer: In this work, all CIPS samples were mechanically exfoliated from a single crystal. We found that 50 nm-thick CIPS flakes could be obtained with good reproducibility. In contrast, consistently obtaining thinner samples of the same thickness is relatively more challenging. Moreover, in the case of a thicker sample of 50 nm, a small thickness variation of ~ 1 nm had a negligible impact on the experimental results and device performance. These factors make 50 nm-thick CIPS a practical and stable choice, particularly in terms of reproducibility.

Figure R1. AFM images of a thinner CIPS sample on Ti substrate, the line height profile indicates the thickness of ~ 24 nm.

In addition to the 50 nm-thick samples, we also conducted mechanical switching experiments on a 24 nm-thick CIPS flake placed on a Ti substrate as shown in Figure R1. Figure R2 presents the PFM phase images of the 24 nm-thick CIPS/Ti sample after initialization and following two sequential 4 ms-long force pulses of 300 nN and 400 nN. Notably, a weaker force pulse of 200 nN was insufficient to induce polarization switching. A single 4 ms-long pulse of 300 nN resulted in 18.7% of the area switching to the opposite

polarization, driven by the inhomogeneous nucleation of downward-polarized domains, thereby confirming mechanical switching. Increasing the applied force to 400 nN further expanded the switched area to 56.2%. Based on these results, the threshold force per unit thickness for the 24 nm-thick CIPS sample is estimated to be ~ 12.5 nN/nm, which is in good agreement with the value of ~ 12 nN/nm obtained from the 50 nm-thick sample.

Figure R2. PFM phase images of the 24 nm-thick CIPS on Ti obtained after initialization and after sequential force applications of 4 ms-long 300 nN and 400 nN. Size: $1 \mu\text{m} \times 1 \mu\text{m}$. In particular, a single 4 ms-long force pulse of 300 nN results in the formation of oppositely polarized domains, covering 18.7% of the area, confirming mechanical polarization switching. A stronger force of 400 nN further increases the switched domain area to 56.2%.

We have added the following discussion in the revised manuscript.

“Here, 50 nm-thick CIPS samples were employed, as they were more easily obtained by mechanical exfoliation compared to thinner flakes, while offering good reproducibility and negligible thickness variation.”

“The low threshold force per unit thickness for mechanical switching was further verified using a thinner 24 nm-thick CIPS sample (Fig. S5). A single 4 ms-long force pulse of 300 nN was sufficient to induce mechanical polarization switching (Fig. S6), corresponding to a threshold force per unit thickness of 12.5 nN/nm, which is in good agreement with the ~ 12 nN/nm value obtained from the 50 nm-thick sample.”

3. How does the application of three sequential force pulses to a transistor result in different conductance states? The physics behind seems poorly demonstrated.

Answer: We apologize for the missing explanation regarding the conductance states

controlled by sequential force applications. In general, the conductance of the MoS₂ channel is modulated by the ferroelectric polarization state of the overlying CIPS layer, analogous to the operation of a conventional ferroelectric transistor. When the CIPS layer is polarized upward, negative polarization charges aggregate over the CIPS-MoS₂ interface and repulse electrons at the channel. In other words, the resulting polarization field shifts the Fermi level of the n-type MoS₂ closer to the valence band, leading to a high-resistance state. Conversely, when the CIPS layer is polarized downward, the Fermi level of MoS₂ is raised toward the conduction band, enhancing carrier transport and yielding a low-resistance (high-conductance) state.

In our measurements, the CIPS layer was initialized to the upward polarization state prior to the application of force pulses, and thus the MoS₂ channel exhibited a high-resistance state. The application of mechanical force pulses induces a flexoelectric effect, which gradually switches the CIPS polarization to the downward direction. The gradual polarization switching induced by sequential force pulses is visualized in Figure 3a. The regions of MoS₂ beneath these down-polarized CIPS domains are in the low-resistance state. Therefore, increasing the number of applied force pulses gradually increases the ratio of down-polarized domains within the CIPS layer, leading to a corresponding enlarged high-conductance MoS₂ regions. As a result, the overall device conductance gradually evolves from a low-conductance state to higher conductance states upon the sequential application of mechanical force pulses. In the conventional electrically-tuned FeFET, similar multi-level conductance states can also be achieved by controlling gradual polarization switching through the application of fast gate voltage pulses [Chip, 2(2), 100044 (2023)].

We have added the following statement in the revised manuscript to clarify the multi-conductance states achieved by sequential force applications.

“The Fermi level of MoS₂ is pushed toward the valence band under the up-polarized region, which blocks the carrier transport in MoS₂, resulting in a high-resistance state with a conductance of 0.035 nS.”

“Subsequently, the Fermi level is shifted upward in MoS₂ by the downward polarization.”

“The CIPS layer was initialized to an upward polarization state prior to the application of mechanical force pulses, thereby setting the MoS₂ channel in a high-resistance state.

Subsequently, three sequential force pulses were applied via mechanical gating. Each pulse gradually switches the CIPS polarization toward the downward direction as illustrated in Fig. 3a. The regions of MoS₂ beneath the down-polarized CIPS domains exhibit a low-resistance state. Accordingly, increasing the number of applied force pulses progressively expands the down-polarized domains in the CIPS layer, resulting in a corresponding increase in the high-conductance regions within the MoS₂ channel. As a result, by applying three sequential force pulses, we demonstrate four distinct conductance levels, corresponding to a 2-bit data stream from “00” to “11” (Fig. 4c). Similar multi-level conductance states have been achieved in conventional ferroelectric field-effect transistors, where gradual polarization switching was controlled through the application of gate voltage pulses.”

4. How/does the imprint field change by the metal type and CIPS flake dimensions?

Answer: The imprint field in CIPS is influenced by both the electrode metal type and the thickness of the ferroelectric flake through multiple interrelated mechanisms. As discussed in our manuscript, the built-in imprint field primarily arises from the difference between the metal work function and the electron affinity of CIPS. In our experiments, the top electrode is fixed as platinum Pt, which has a higher work function of ~5.65 eV than CIPS (~4.80 eV). This difference drives electron transfer from CIPS to Pt, resulting in hole accumulation at the CIPS/Pt interface and a downward-pointing imprint field. When the bottom electrode is Au (5.20 eV), the work function difference with Pt is smaller, leading to reduced hole accumulation and a weaker imprint field. Replacing the bottom electrode with low work function metal Ti (4.33 eV) causes electron transferred to CIPS, enhancing electron accumulation and generating a stronger imprint field pointing from Pt to Ti. Beyond work function alignment, the metal type also influences the imprint field through interfacial charge screening, chemical interactions, and defect dynamics. In addition to metal type, the thickness of the ferroelectric also affects the imprint field. Simply speaking, the imprint field in ferroelectrics is inversely related to thickness within the functional ferroelectric range, driven by the combination of interfacial phenomena and size-dependent electrostatic effects. Overall, achieving control of the imprint field is complex, as it requires comprehensive optimization of both the electrode material properties and the ferroelectric thickness, considering the

interplay of work function differences, charge screening, interfacial chemistry, and size-dependent electrostatic effects.

We have added the following discussion in the revised manuscript to reflect this issue.

“Achieving precise control of the imprint field is complex, as it requires comprehensive optimization of both the electrode material properties and the ferroelectric thickness, considering the interplay of work function differences, charge screening, interfacial chemistry, and size-dependent electrostatic effects.”

5. How does the grain boundary of CIPS affect the switching properties? Authors should briefly describe it in the manuscript.

Answer: Grain boundaries can influence ferroelectric switching behavior by acting as pinning centers that hinder domain wall motion, thereby affecting the switching dynamics and stability of the material. In this work, the single crystal thin CIPS samples were employed from exfoliating from a high quality CIPS single crystal. Both the force-induced and voltage-driven switching processes exhibited smooth domain evolution in the PFM phase images, with no evidence showing the existence of grain boundary.

We have added the following discussion regarding this issue in the revised manuscript.

“Then CIPS samples were mechanically exfoliated from a commercially sourced bulk single crystal (Six Carbon Technology, Shenzhen) and transferred onto target substrates. The use of single crystalline samples eliminates undesired effects caused by grain boundaries.”

Reviewer #3 (Remarks to the Author):

The work by Zhang et al. presents the case of CuInP2S6, a two-dimensional (2D) ferroelectric material with spontaneous out-of-plane electric polarization, and its potential use as a flexoelectric material. Their results report the lowest threshold force per thickness by the implementation of a strategy based on rapid, low force pulses to control of the imprint field via asymmetric boundary design with little damage to the sample. They also show the potential applications of the technique in functional gated devices based on van der Waals heterostructures (made purely using stacked 2D materials), where the switch in polarization

in the ferroelectric material is induced by the mechanical strain and not by an external electric field. The manuscript is well written, easy to understand and provides an extensive literature review. As the authors deal with a 2D ferroelectric material, I believe this work can appeal to a broader audience in the field of two-dimensional materials and devices. However, because of this appeal to the condensed matter community in general, I would like the authors to address my concerns before publication:

1. It is clear that the threshold force is a relevant factor when dealing with ferroelectric materials, and trying to minimize it would benefit their implementation in commercial devices. However, I miss a proper definition of what an imprint field is and how it is relevant (it is not very clear to me in the text).

Answer: We are sorry for the insufficient explanation. The imprint field in a ferroelectric material refers to a built-in internal electric field that biases the polarization state, favoring one direction over the other. This field shifts the polarization-electric field hysteresis loop along the electric field axis (as illustrated in Fig. 2b), leading to a preferred polarization direction. The imprint field originates from asymmetries in the device structure, such as differences in work function between the top and bottom electrodes. In our case, the asymmetry in work functions between the top and bottom contacts, e.g., Pt and Ti, leads to an internal electric field across the CIPS layer, which breaks the degeneracy of polarization states. Energetically, energy barrier for switching polarization is reduced in the direction favored by the imprint field. As flexoelectric-induced polarization switching using the probe tip is unidirectional, the imprint field can be designed with the asymmetric boundary and induced to lower the energy barrier of flexoelectric-induced polarization switching. Therefore, a favorable imprint field can lower the threshold force required for mechanical switching.

We have added the following discussion to the revised manuscript.

“An imprint field in a ferroelectric material is an internal electric field that biases the polarization state, favoring one direction over the other. In ferroelectric devices, imprint fields can have both detrimental and beneficial effects. On one hand, they can lead to asymmetric switching behavior and retention loss in the energetically disfavored polarization state. On the other hand, they can be engineered to guide a desired polarization direction by

reducing the energy barrier. In terms of flexoelectric-induced polarization switching, an imprint field aligned with the flexoelectric field can effectively lower the energy barrier for one polarization direction ($\Delta E_2 < \Delta E_1$), enabling switching at reduced mechanical force. Such imprint field can arise from structural asymmetries, e.g., differences in work function between the top and bottom electrodes. Achieving precise control of the imprint field is complex, as it requires comprehensive optimization of both the electrode material properties and the ferroelectric thickness, considering the interplay of work function differences, charge screening, interfacial chemistry, and size-dependent electrostatic effects. Nevertheless, by contacting the ferroelectric surfaces with selected materials, an imprint field can be introduced to facilitate flexoelectric polarization switching.”

2. I found in the literature that having zero imprint field benefits the functionality of ferroelectric-based devices [Sci. Rep. 6, 25028 (2016)], but the authors claim that this imprint field is important for allowing a relatively easy polarization switching. I find both statements contradictory, so I would like the authors to comment on this and clarify it.

Answer: The imprint field in ferroelectric devices can have both beneficial and detrimental effects. On the downside, it degrades device performance by causing asymmetric switching and retention loss in the unfavored polarization state. For instance, in a conventional ferroelectric transistor where balanced voltages for writing and erasing operations are desired, a zero-imprint field is often preferred as discussed in [Sci. Rep. 6, 25028 (2016)]. On the upside, the imprint field can be intentionally engineered to stabilize or reduce switching barrier for a preferred polarization direction at lowered energy [for instance, ACS Appl. Electron. Mater. 5, 4615-4623 (2023); J. Mater. Chem. C, 12, 15188-15200 (2024); Adv. Funct. Mater. 2502700 (2025)]. In our work, we are particularly concerned with minimizing the mechanical force required for polarization switching, in order to avoid damage to the CIPS material. Hence, we take advantage of the designed imprint field to lower the switching barrier for flexoelectric-induced polarization switching to facilitate the low force operation. Although the voltage required for upward polarization increases slightly, the reduction in threshold force of mechanical switching benefit with more reliable mechanical operation. Therefore, in our device design, a non-zero imprint field plays a beneficial role in enabling

low-force polarization switching, which is crucial for the integrity and durability of the ferroelectric material.

We have added the following discussion to clarify on this issue.

“In ferroelectric devices, imprint fields can have both detrimental and beneficial effects. On one hand, they can lead to asymmetric switching behavior and retention loss in the energetically disfavored polarization state. On the other hand, they can be engineered to guide a desired polarization direction by reducing the energy barrier. In terms of flexoelectric-induced polarization switching, an imprint field aligned with the flexoelectric field can effectively lower the energy barrier for one polarization direction ($\Delta E_2 < \Delta E_1$), enabling switching at reduced mechanical force.”

3. I find appealing the application of the described technique using van der Waals heterostructures, as it is known that applying high electric fields in CIPS can induce the migration. Did the authors just measure one sample? From the text, I assume only one device was prepared and tested (as shown in Figure S7). When dealing with devices, it is very important to test the reproducibility of the observed phenomena in more than one sample, so I strongly recommend to at least measure another with the same configuration proving that the several samples show the same gate modulation when applying the force pulses. I would also like to see the image of the device as part of Figure 4, as I think it provides a visual representation of the experiment.

Answer: Thank you for raising this important issue. The ferroionic effect with server Cu^+ immigration can be induced in CIPS by applying high static voltage [Nat. Commun. 13, 574 (2022)]. In our experiment, a fast negative voltage pulse of -4 V was sufficient and was employed to reserve the polarization. With the short voltage application, we did not notice any evidence of ferroionic effect, which normally causes topographic change or even damage associated with the immigration of Cu^+ ions [Science Advances, 8, abq1232 (2022); Nat. Commun. 13, 574 (2022)]. For instance, as shown in Fig. 3c, the smooth surface is maintained after two cycles of mechanical writing and electrical erasing.

We have added the following discussion regarding this issue in the revised manuscript.

“Prior to mechanical stimulation, a fast pulse of negative voltage of -4 V was applied to

initialize the measured area to an upward polarization state while preventing material damage from ferroionic effect.”

Figure R3. Optical microscope image of the fabricated transistor presented in Fig.4c. The dashed lines provide the eye guides to the edges of each functional layers.

We have measured and demonstrated two mechanically-gated transistors in the previous manuscript. Fig. 4b and Fig. 4c were indeed measured in two different devices. However, we only included the microscope image for the device of Fig. 4b in SI. In the revised SI, we have added the device image for Fig. 4c (see Figure R3) as Fig. S8.

“The optical microscope image of the measured device is presented in Fig. S8.”

Figure R4. Revised Fig. 4a including the optical microscope image of a fabricated device. Optical microscope image of a fabricated transistor. The dashed lines provide the eye guides to the edges of each functional layers. Scale bar: 5 μm.

Furthermore, as suggested by the review, the microscope image of a device has been included in the revised Fig. 4 for a better visual presentation as shown in Figure. R4.

“The optical microscope image of a fabricated device is presented in Fig. 4a.”

Figure R5. Optical microscope image of the additional device fabricated and measured to verify the reproducibility of multi-level conductance states controlled by mechanical gate.

Figure R6. Multi-level conductance behavior controlled by force pulses via the mechanical gate in the device shown in Figure R5. The output curves demonstrate three distinct conductance states induced by sequential gate pulse operations. The retention characteristics of these states are maintained for at least 200 seconds. Insets show the corresponding PFM phase images of the gated CIPS region on the top the MoS₂ channel, as a rectangular area between graphene contacts along the current path. Gradual polarization switching is clearly observed after each gate force pulse application, supporting the reproducibility of the conductance states.

To further validate the reproducibility of the multi-level conductance feature controlled

by the mechanical gate, we fabricated a third device with the same configuration shown in Figure R5. Figure R6 presents the output curves of the transistor under gate pulse operations, revealing three distinct conductance states. Retention test also demonstrates three conductance states measured up to 200 seconds. PFM phase images were also measured in the gated region of the CIPS layer on top of the MoS₂ (a rectangular area between graphene contacts) under each conductance state. Clear gradual polarization switching is noted after each gate force pulse application. These results provide further evidence for the reproducibility of the multi-level conductance states controlled by applying force pulses at the mechanical gate.

Figures R5 and R6 have been added in the revised SI as Fig. S9 and Fig. S10. The following statement has been added in the revised manuscript accordingly.

“An additional device was fabricated and measured to verify the reproducibility of the multi-level conductance states controlled by applied force pulses (Fig. S9). Three distinct, nonvolatile conductance states were demonstrated by gradually switching the polarization of CIPS through sequential force pulses (Fig. S10).”

4. I think it is important to specify the source of the CIPS crystals (can be written in the methods section); were they synthesized by the authors, or were they obtained from a company? Same goes for the MoS₂ and HOPG.

Answer: We are sorry for the missing details. All materials used in this work are exfoliated from commercially sourced bulk single crystals. CIPS was brought from Six Carbon Technology Shenzhen, MoS₂ and HOPG were brought from HQ Graphene Groningen.

We have added the information in the revised Method section.

“Then CIPS samples were mechanically exfoliated from a commercially sourced bulk single crystal (Six Carbon Technology, Shenzhen) and transferred onto target substrates.”

“Graphene flakes were mechanically exfoliated from single-crystal HOPG (HQ Graphene Groningen) and transferred onto a SiO₂/Si substrate.”

“A few-layer MoS₂ flake was exfoliated from the single crystal (HQ Graphene Groningen).”

5. Please, write the units of the colour scale in Figure S8.

Answer: Sorry for the ambiguity. Fig. S12 (Fig. S8 in previous version) shows the distribution of strain, a dimensionless quantity that represents relative deformation, i.e., the ratio of the change in length to the initial length.

We have revised the caption of Fig. S12 to avoid the ambiguity.

“ e_{xx} , e_{yy} and e_{zz} are the strain distributions along various directions caused by the tip-induced deformation, which are dimensionless quantities representing relative deformation, i.e., the ratio of the change in length to the initial length.”

6. Just a curiosity: did the authors choose CIPS instead of other layered ferroelectric materials (i.e. In_2Se_3) for any particular reason?

Answer: To prototype a mechanically gated transistor, the ferroelectric material should function as an insulator. CIPS is the most stable, well-established and commercially available ferroelectric insulator, exhibiting significant out-of-plane ferroelectricity. This makes it an ideal choice as a ferroelectric dielectric layer in the application of ferroelectric transistor. In contrast, In_2Se_3 is a ferroelectric semiconductor, which limits its suitability as a gate dielectric. Furthermore, a pronounced flexoelectric effect has been reported in CIPS.

If the authors address my concerns (specially point 3, which I believe is the most exciting part of the work if developed further), I can recommend the publication of the manuscript in Nature Communications. If it is not possible to prove the claim regarding the devices providing more samples, I would suggest to transfer the article to npj Flexible Electronics as I think it fits better with the ‘Aims & Scope’ of the journal.